## Overview Review

forced displacement; health economics; conflict; global mental health policy; global mental health

**Author for correspondence:**
David McDaid,
Email: d.mcdaid@lse.ac.uk

# Making an economic argument for investment in global mental health: The case of conflict-affected refugees and displaced people

David McDaid and A-La Park

Care Policy and Evaluation Centre, Department of Health Policy, London School of Economics and Political Science, London, UK

## Abstract

Mental health expenditure accounts for just 2.1% of total domestic governmental health expenditure per capita. There is an economic, as well as moral, imperative to invest more in mental health given the long-term adverse impacts of mental disorders. This paper focuses on how economic evidence can be used to support the case for action on global mental health, focusing on refugees and people displaced due to conflict. Refugees present almost unique challenges as some policy makers may be reluctant to divert scarce resources away from the domestic population to these population groups. A rapid systematic scoping review was also undertaken to identify economic evaluations of mental health-related interventions for refugees and displaced people and to look at how this evidence base can be strengthened. Only 11 economic evaluations focused on the mental health of refugees, asylum seekers and other displaced people were identified. All but two of these intervention studies potentially could be cost-effective, but only five studies reported cost per quality-adjusted life year gained, a metric allowing the economic case for investment in refugee mental health to be compared with any other health-focused intervention. There is a need for more consistent collection of data on quality of life and the longer-term impacts of intervention. The perspective adopted in economic evaluations may also need broadening to include intersectoral benefits beyond health, as well as identifying complementary benefits to host communities. More use can be also made of modelling, drawing on existing evidence on the effectiveness and resource requirements of interventions delivered in comparable settings to expand the current evidence base. The budgetary impact of any proposed strategy should be considered; modelling could also be used to look at how implementation might be adapted to contain costs and take account of local contextual factors.

## Impact Statement

Spending on mental health globally remains extremely low despite increased visibility, but there are some areas of mental health that are particularly neglected. One pressing issue in the global mental health community is the increase in people fleeing their homes because of war and other conflict. This leads to increased rates of mental health conditions, such as depression, anxiety and traumatic stress disorders. There is also ample evidence of the profound long-term adverse consequences of poor mental health to refugees, asylum seekers and other displaced people, including increased costs to health services for both mental and physical health needs and reduced participation in work and education. Early intervention is needed to identify and provide appropriate support for mental health needs, but access to mental health services is fragmented and limited, even in high-income countries. Economic evidence is widely used to make the case for investment in screening and treatment of communicable diseases among displaced people; in contrast we identified very few economic studies looking at the case for mental health health for refugees, asylum seekers and other displaced people. We highlight this evidence and its limitations and make recommendations on how it can be expanded. This is vital if the economic case is going to change mindsets among policy makers who see the mental health of the displaced as a low priority. We call for more focus on long-term and not just short-term costs and benefits and also to make policy makers aware of any potential additional positive impacts for their domestic populations that could also be achieved. These benefits could include extending access or replicating new services to support local people with similar psychological needs, e.g. health and other frontline workers affected by the pandemic or people affected by natural disasters such as flooding.

## Introduction

More than 20 years ago, the World Health Report first focused on mental health, with then World Health Organization (WHO) Director-General Gro Harlem Brundtland noting that it had been 'neglected for far too long – is crucial to the overall well-being of individuals, societies and countries and must be universally regarded in a new light' (World Health Organization, 2001). The subsequent decades have seen an increase in the visibility of mental health as part of the global health agenda, with a recognition that more must be done to invest in mental health (World Health Organization, 2022).

Despite this increased recognition, investments in the promotion of good mental health, as well as in the prevention and management of mental health conditions have remained low; in high-income countries (HIC) mental health expenditure typically accounts for 3.8%, while in low and middle-income countries (LMICs) it is just 1% to 1.5%, of total domestic governmental health expenditure per capita (World Health Organization, 2020).

This paper reflects on how economic evidence can be used to strengthen the case for action on global mental health, using the specific example of interventions to support populations that have been displaced due to conflict, fear of persecution and human rights violations. This appears to be an area with less economic evidence than other areas of global mental health, and it is a particularly pressing global mental health issue (Patanè et al., 2022). It is also truly global as conflicts have required responses to support the displaced in HICs as well as in LMICs.

By 2021, 89.3 million people had been forcibly displaced from their homes, doubling the number a decade earlier (United Nations High Commission for Refugees, 2022). 60% were internally displaced within their own countries, but the number of refugees fleeing to other countries has been increasing. 83% of refugees are hosted in LMICs, with Türkiye hosting 3.8 million, followed by Colombia with a further 1.8 million. This has been compounded by the conflict in Ukraine; by July 2022 5.2 million Ukrainians had sought refuge in other European countries (UNHCR, 2022a), with a further 7 million internally displaced (UNHCR, 2022b). There also has been an increase in displacement and economic hardship following adverse weather and natural disasters, something that may become even more of an issue due to continuing climate change (Hoffmann et al., 2020; Charlson et al., 2022).

Arguments illustrating that there is an economic, as well as moral and public health, imperative to invest more in global mental health may help facilitate implementation. This paper looks at how the economic case for investment in measures to support the mental health of refugees and displaced people can be strengthened to inform policy making. It focuses on three fundamental interlinked economic questions used to inform global mental health policy and practice: (i) what are the costs of not taking action on mental health, (ii) what are the costs of taking action and finally (iii) what is the cost-effectiveness of these actions?

Evidence on the economic costs of poor mental health is highlighted and the existing economic evaluation evidence-base considered. Barriers and facilitators to the use of economic evidence for this population group are set out, with an emphasis on making an economic case beyond the direct health and welfare benefits to displaced populations that also considers wider impacts to the economy to directly appeal to both health and non-health sector system budget holders.

## Using economic arguments to facilitate investment

### Quantifying the costs of not taking action

Not intervening to support global mental health is not costless; in addition to immediate adverse impacts on global mental health there are wider longer-term impacts to society, indeed the majority of the economic costs of poor mental health manifest themselves in reduced participation in economic activities, such as employment, education, volunteering and family caring (Christensen et al., 2020).

As part of any economic argument, these short and long-term impacts need to be identified and valued monetarily where feasible. This could be done by collating what is now compelling evidence of the long-term adverse mental and physical health consequences, not just from the trauma experienced by refugees, but also from the ways in which they are welcomed and integrated in relevant host country settings, including their access to routine health care and welfare services. A previous systematic review of HICs found multiple studies associating poor mental health outcomes with restrictions on immigration entry, as well as subsequent restrictions on access to employment and education (Juárez et al., 2019). More understanding about these factors, could help strengthen the case for early identification and intervention for at-risk individuals to avoid some of the long-term consequences to health systems, both mental and physical.

Evidence from systematic reviews indicates higher rates of depression and PTSD seen in refugees can last for many years, suggesting the need for long-term support systems (Blackmore et al., 2020; Hoell et al., 2021). For children especially there may be profound impacts on future life chances if schooling becomes further disrupted because of unaddressed depression, anxiety and PTSD (Blackmore et al., 2020). There are also physical health consequences, for example, higher rates of cardiovascular disease and hypertension in Vietnamese and Cambodian refugees in the US who still experience trauma due to conflict many decades later (Wagner et al., 2013).

Appropriate local estimates of the costs of health and other services could be combined with information on the increased use of services from literature to give policy makers some sense of the magnitude of the costs of doing nothing. Local data could also be collected as part of evaluations on both short, and where feasible, longer-term outcomes. Standard templates for asking questions about the use of health and services, as well as other impacts including time out of work or the need for informal care, such as the Client Service Receipt Inventory (CSRI), have been adapted to collect information from refugees and displaced people (McCrone et al., 2005; Grochtdreis et al., 2021; Acarturk et al., 2022; Spaaij et al., 2022). They can also be used to compare wait times, out of pocket costs and time spent receiving health care and other services compared to local populations. In host countries, it might also be feasible to conduct (or add questions to) general population surveys on longer-term health and social outcomes, such as participation in employment and level of earnings. In some HICs, electronic health and welfare record data have also been accessed (Maier et al., 2010). Impacts on families could also be measured through surveys and electronic service records (Bager et al., 2018). All of these approaches can be seen in published costing studies, but this evidence base is limited and mainly focused on the experience of refugees living in HICs.

For example, interviews with 78 asylum seekers in Switzerland, as well as the use of electronic health records, suggest that annual health care costs for refugees are almost double those for the Swiss

general population, with a key potential driver being undiagnosed mental disorders; moreover the costs for refugees with a mental health condition may be even greater than for refugees without these conditions (Maier et al., 2010). Longer-term analysis in Switzerland using health insurance system data also indicates asylum seekers with mental health conditions have increasing health care costs over time compared to asylum seekers without these conditions. A lack of early intervention to identify and refer asylum seekers to appropriate services may contribute to this increasing cost trajectory (Tzogiou et al., 2022).

In Germany, initial annual mental health service use and general health service use for Syrian refugees with untreated PTSD were much lower than for the local population with similar mental health needs, potentially because of barriers such as mental health stigma, language, coverage restrictions and a lack of awareness of services among refugees (Grochtdreis et al., 2021). At the same time, another German study indicated rates of hospitalisation due to mental health conditions in asylum seekers were twice as high as those in the general insured population (Bauhoff and Göpffarth, 2018); that study also suggested increased access to primary care led mental health services would reduce costs.

There is some limited evidence on the costs of no intervention from LMICs. For example, interviews with 264 people from the former Yugoslavia who experienced conflict-related PTSD but never received treatment revealed that, on average 11 years later, they still had high levels of PTSD and incurred substantial healthcare-related costs, particularly a need for unpaid informal family care (Priebe et al., 2009). Another study on the Yugoslav conflict also suggests an association between increased long-term health service utilisation/costs (on average 8 years after exposure to conflict) and depression, anxiety and/or PTSD, regardless of whether still living in the Balkans or resettling in western Europe (Sabes-Figuera et al., 2012). A cross-sectional survey in Kosovo reported very low utilisation of mental health services by internally displaced or refugee women during the Kosovan war (Morina and Emmelkamp, 2012); the lack of good quality mental health services may have influenced service utilisation. Elsewhere, other analyses looking at Somali refugees in the UK (McCrone et al., 2005), Syrian refugees in Turkey (Fuhr et al., 2019; Acarturk et al., 2022) and internally displaced people in Georgia (Chikovani et al., 2015) all report low levels of contact with mental health services despite mental health needs. While economic evidence on the association between restricted access to health services and longer-term mental health outcomes appears limited in LMICS, reduced, potentially inadequate, levels of contact with services will occur when out of pocket costs and other barriers to service use are introduced, as for instance seen in Jordan after the costs of access to health services for Syrian refugees rose (Abu Siam and Rubio Gómez, 2021). These examples of low utilisation of services potentially could increase future risk of more intensive need for health service use.

### Quantifying the costs of taking action

It is not enough to identify the costs of not taking action, evidence on effective interventions to mitigate these impacts in different global settings are needed. However, even if effective interventions are available, there will always be difficult choices to be made about the way in which limited resources can be used. An essential precursor to any assessment of cost-effectiveness is to measure resources required to implement any mental health intervention. As part of the evaluation, this means documenting all aspects of intervention implementation, including training and initial set-up costs, time of frontline staff delivering services, any in-kind support received such as free use of venues and the time of volunteers. Travel and waiting time, as well as incidental costs incurred by service recipients and any accompanying family members to receive any intervention, should also be recorded. Ideally, this should be done prospectively as part of evaluation; if this information has not been collected an alternative is to have a subsequent discussion with programme implementers to describe treatment pathways and estimate typical resources required for each step along the care pathway. Once all resources have been identified appropriate locally sourced costs can be attached, such as hourly rates for different health care professionals or a minimum wage or replacement cost value for unpaid volunteer and family care time. Some of this information on resource use and cost may be in routinely prepared financial reports; this can be combined with information on the number of service recipients to generate some information on implementation costs (Jordans et al., 2011).

Tools have also been developed to aid evaluators and decision-makers in quantifying some of these costs. In Europe, for example, standardised unit costs for services for health, social care, and some other sectors including educational and criminal justice are available (Mayer et al., 2022). Other tools have been developed that focus on LMICs (Jeet et al., 2021). They include the OneHealth Tool, a freely available online template which can be used by evaluators and policy makers to estimate resource requirements and costs associated with differing levels of population coverage for different health policy strategies combining one or more elements of packages of care (Chisholm et al., 2017).

### Carrying out an economic evaluation

Having quantified the costs of doing nothing and estimated the economic value of interventions, policy makers still need to know whether this represents a good use of resources. This means that in addition to determining whether strategies are effective, and quantifying how much they cost, it is important to determine opportunity costs associated with any impacts, as there are always alternative ways in which resources could be used. This is where economic evaluation, involving an analysis of costs and outcomes achieved from at least one intervention, compared with one or more alternative courses of action, or no action, can be used to help inform policy-making decision on the provision of services.

In many HICs economic evaluation is an integral part of the policy-making process within health systems, helping to determine which interventions, particularly pharmaceuticals, are funded and their eligible populations. Although less used in LMICs, international agencies such as the WHO have promoted the increased use of economic evidence in decision-making, for both country-level policy makers and international donor agencies, for instance highlighting the economic case for tackling depression and anxiety (Chisholm et al., 2016).

### Approaches to economic evaluation

There are several different ways in which economic evaluations can be conducted and detailed guidance is available (Drummond et al., 2015); in brief, the main approaches all measure costs similarly, but differ in how outcomes are measured. Cost-effectiveness analyses (CEA) measure changes in one or more specific outcome measures, for example, changes in the number of depression-free days. This is of limited use if decision-makers want to compare investments in

depression mitigation with completely different health conditions, for example, rates of communicable disease. Cost-utility analyses (CUA) overcome this issue by the use of common outcome metric for all health conditions.

One measure used in CUA is the quality-adjusted life year (QALY). This combines information on both the quantity and quality of life lived where a value of 1 represents a year spent in perfect quality health and 0 represents a year in the worst imaginable health condition (usually death). Quality of Life can be elicited in different ways, for instance asking study respondents to complete the EQ-5D (Devlin and Brooks, 2017), an instrument often used in health economic studies. It is available in many languages and information on values for each possible health state in some different HICs and LMICs is available. The sensitivity of the EQ-5D to some mental health conditions has been questioned, and an alternative instrument specifically designed to capture quality of life in economic analyses in people living with mental health conditions, the Recovering Quality of Life (ReQoL) measure has recently been developed (Keetharuth et al., 2018). Regardless of which measure is used QALYs can be estimated. Another approach, for example, transforms World Health Organization Disability Assessment Schedule scale (WHO-DAS 2.0) scores into utility values (Lokkerbol et al., 2021). Once done QALYs can be calculated; for example, if individuals indicate that the utility associated with their health state, for example, moderate depression, is 0.8 and they spend 5 years living in this health state this equates to 4 QALYs (0.8 × 5 years).

Many economic evaluations in LMICs use the disability adjusted life year (DALY), this reflects the sum of time lost due to premature mortality and years lived with different levels of disability. The disability weights used for DALYs, which have been determined by expert opinion, are the inverse of those for utility, with '0' referring to no disability and '1' representing being dead. Although it is conceptually different, one recent review found that differences in the relative ratio between costs and outcomes in studies using QALYs or DALYs are usually modest, so the choice in any context between using QALYs or DALYs may be 'unlikely to materially affect resource allocation recommendations' (Feng et al., 2020).

Both CEA and CUA still leave policy makers having to make subjective judgements when an intervention has better outcomes but costs more than the alternative options. The threshold at which an intervention is still considered cost-effective will vary between different country contexts. For example, the threshold in England used in most decisions on access to new interventions is a cost per QALY gained of no more than £20,000 to £30,000 (National Institute for Health and Care Excellence, 2022), approximately equivalent to 0.66 to 0.99 of Gross Domestic Product (GDP) per capita. Cost per DALY averted thresholds used in studies have varied from 0.34 of GDP per capita in low human development index (HDI) countries, compared with 0.67, 1.22 and 1.46 of GDP per capita in medium HDI, high HDI and very high HDI countries (Daroudi et al., 2021).

Most CEA or CUA studies only look at health-related outcomes. Interventions that improve mental health will have wider benefits for other sectors that policy makers may also be interested in, for instance, better mental health has been associated lower risk of contact with the criminal justice system and better educational attainment. Mental health interventions may also be funded outside of the health sector, for example, some refugee mental health interventions may be funded by international donor organisations who will have different priorities, including economic recovery and social integration, better opportunities for women and girls and reduction of gender-based violence.

The third main approach to economic evaluation, cost-benefit analysis (CBA), is especially useful in this context, as it can place a monetary value on all outcomes. The decision rule is also simpler: an intervention is considered a good investment if the value of benefits outweighs the costs incurred. Again, different methods can be used to identify monetary value of outcomes, including surveys asking respondents how much they are willing to pay for a specific outcome. A monetary value can also be placed on a QALY or DALY to generate a net monetary benefit, but this varies across countries, reflecting differences in societal willingness to pay for a QALY.

A related concept is return on investment (ROI) analysis. This compares the costs of any action with any costs that may be avoided as a result of favourable outcomes and is being used to help make the long-term case for global mental health interventions (Chisholm et al., 2016). This ROI approach can also be broken down to show how the return changes over different time periods, indicating which sectors of economy benefit. It can help demonstrate that overall to government there may be a positive case for investment which would not be apparent if looking at impacts on one sector alone (McDaid and Park, 2016).

## Use of economic evaluation for mental-health-related interventions for refugees and displaced people

We undertook a rapid systematic scoping review looking specifically for economic evaluations to better manage the mental health of refugees and displaced populations. While the focus was on populations from LMICs, these interventions could be delivered in any setting, including high-income host countries. The search covered PubMed, PsychINFO, Embase, Global Health, PAHO, Desastre-Disasters Latin America and Caribbean Literature in Health Sciences (LILACS) and the Índice Bibliográfico Español en Ciencias de la Salud) (IBECS) databases from inception to July 2022 (see Supplementary Material for search strategies used), as well as a limited search of Google Scholar. In addition, we looked for any economic analysis reported in existing efficacy and effectiveness reviews of evaluations of mental health-related interventions for refugees and displaced people.

From 530 titles/abstracts initially retrieved only 11 economic evaluations were identified (Table 1), suggesting that despite the grave situation facing displaced populations this is an area of global mental health where few economic evaluations have been conducted. This is in line with other reviews. A review of interventions for the management of conflict-related trauma in LMICs was unable to find any economic evaluation (Al-Tamimi and Leavey, 2022). We also identified a review of economic evaluations and costing studies up to February 2020 for people with post-traumatic stress disorder (PTSD) (von der Warth et al., 2020). It included 18 economic evaluations, but only one was concerned with the mental health of a conflict-affected population (Chang et al., 2018). These findings also contrast with economic evidence for communicable disease in refugees, in our review we excluded numerous studies looking to screen, vaccinate and treat various communicable diseases.

We were only able to find one study that looked at the economic case for mental health screening for refugees and displaced populations. An exploratory decision-analytical modelling study in Germany estimated the incremental costs per QALY gained for universal screening followed by CBT treatment for mild and

**Table 1.** Summary of economic evaluations on mental-health interventions for refugees and internally displaced people

| References Country of study | Setting and study population (age, sex, size) | Intervention details (study design, description of intervention and comparator) | Economic analysis | | Outcomes and key findings | | |
|---|---|---|---|---|---|---|---|
| | | | Perspective Price Year Currency | Type of economic analysis Study duration | Effect on mental health | Resource and cost impacts | Key economic findings |
| Bager et al., 2018 Denmark | 45 adult refugees (66% Iraqi) who have been severely traumatised as a result of torture. 44 matched controls for long-term CBA | Observational study. Intervention: Multi-disciplinary rehabilitation programme for severely traumatised torture survivors. Mean 14.3 months of treatment Control: refugees currently on waiting list for treatment only for CBA | Societal 2016 Danish Kroner (DKK) | CUA and CBA 23 months (CUA) 14 years (CBA) | Transformation of WHOQOL-Brief to obtain QALYs. 0.82 mean QALY gain over 23 months | Mean costs of treatment only DKK 166,113 (2001 prices) Registry data used to measure employment participation and health service use (No breakdown provided) | Cost per QALY gained DKK 262,530 (Authors note value within UK cost-effectiveness threshold) In CBA from individual perspective after 14 years costs outweigh benefits. From family perspective positive net benefits after 3 years |
| Barron et al., 2016 Palestinian Territory | 154 adolescents affected by conflict and meeting criteria for PTSD using the CRIES-8 instrument | RCT. Intervention: 5 sessions of Teaching Recovery Techniques Cognitive Group Behavioural Intervention. Control: wait list receiving usual social education curriculum | External donor/funder Not stated US Dollars ($) | CCA Not stated – by 2 weeks after end of intervention | Significantly more young people in intervention group no longer met criteria for PTSD | Full cost of intervention including training, and expenses for children was $38.68 | No formal synthesis. Authors state costs to be modest relative to benefits to children |
| Biddle et al., 2019 Germany | Hypothetical population of 1,000 newly arrived adult refugees and asylum seekers | Modelling study. Intervention: Nurse screening using PHQ-9 for depression with 12 sessions of CBT over 3 months for moderate/severe depression. Comparator: Case-finding via self-referrals and follow-up care by non-profit psychosocial centres | Health system 2017 Euros (€) | CUA 15 months | For 1,000 cohort gain of 2.95 QALYs for screening versus case finding (EQ-5D) mean gain 0.003 QALYs) | Mean costs of screening €2.75. Total costs of diagnosis, treatment, and health system costs of depression €137,398 and € 78,982 for intervention and comparator groups | Cost per QALY gained €19,779. 83% chance of cost per QALY gained below €50,000 |
| Böge et al., 2022 Germany | 584 Arabic or Farsi-speaking refugees and asylum seekers aged 14–65 with at least mild depressive symptoms (PHQ-9 or PHQ-Adolescents score of 5 or more) and psychological distress (Refugee Health Screener RHS-15) | RCT. Intervention: 4 Level Stepped Care and Collaborative Care Model. Comparator: Treatment as usual. Maximum treatment duration for each level 12 weeks | Health service 2019 Euros (€) | CEA and CUA 1 year | Significant improvements in depressive symptoms and remission. Non-significant mean improvement in QALYs (0.019) in intervention group | Total mean cost of intervention group €1,781, including €312 for intervention versus €1,921 for comparator group. Overall no significant difference but health care costs significantly lower in intervention group ($p = 0.007$) | The intervention was dominant with significantly lower health system costs and (non-significantly) improved QALYs gained. High probably of being cost-effective in uncertainty analysis |

*(Continued)*

**Table 1.** (Continued)

| References Country of study | Setting and study population (age, sex, size) | Intervention details (study design, description of intervention and comparator) | Economic analysis | | Outcomes and key findings | | |
|---|---|---|---|---|---|---|---|
| | | | Perspective Price Year Currency | Type of economic analysis Study duration | Effect on mental health | Resource and cost impacts | Key economic findings |
| Chang et al., 2018 Kosovo | 34 adults who had trauma, depression or anxiety and had been victims of torture and war | Modelling study using data from RCT. Intervention: 10-week multidisciplinary rehabilitation programme: physiotherapy, biofeedback-supported psychotherapy and social support. Control: Wait-list | Societal 2013 Euros (€) | CEA, CUA and ROI 1 year | No significant difference in mental health outcomes but better in intervention group. 0.097 more QALYs gained in intervention group (significance not reported) | Mean cost of intervention €1,019 Mean monthly income increased by 18% to €133 in intervention group. | Cost per QALY gained €10,505 which is not considered cost-effective in Kosovan context. Gains in productivity would have to maintained for 57 months to generate a positive ROI |
| de Graaf et al., 2020, Netherlands | 60 adult Syrian refugees with psychological distress (Kessler Psychological Distress Scale (K10) score > 15) and reduced psychosocial functioning (WHO-DAS 2.0 > 16) | RCT. Intervention: 5 weekly sessions of Problem Management Plus (PM+) plus care as usual. Control: care as usual | Healthcare and Societal 2018 Euros (€) | CEA 3 months | Significant improvement in depression and anxiety symptoms as well as social functioning | Total mean costs of intervention group €917.65 (including €403 for PM+) versus €645.12 in comparator group ($p = 0.368$). Increased health care costs €138 but productivity losses lower €268 | Incremental cost per recovery achieved €5,047 and €2,266 from health system and societal perspectives |
| Hamdani et al., 2020 Pakistan | 340 adults with high levels of psychological distress (score > 2 on GHQ) and functional impairment (WHO-DAS 2.0 > 16) | RCT. Intervention: 5 weekly sessions of Problem Management Plus (PM+) plus enhanced care as usual. Control: enhanced care as usual | Healthcare 2016 Pakistan Rupees (PKR) | CEA 3 months | Significant improvement in depression and anxiety symptoms as well as social functioning | Total mean costs of intervention group PKR 18,074 (including 17,473 for PM+) versus PKR 1,696 in comparator group. No significant differences in health and social care utilisation/costs | Incremental cost per successfully treated case of depression PKR 53,770 but would reduce to PKR 10,705 if local rather than international trainers used |
| McBain et al., 2016 Sierra Leone | 436 people aged 15–24 affected by war, not attending school, with psychological distress half of one standard deviation above that previously seen in war-affected young people in the region (Oxford Measure of Psychosocial Adjustment); emotional and behavioural problems having impacts on day to day functioning | RCT. Intervention: Youth Readiness Intervention (YRI), 10 week multi-component therapy to reduce functional impairment. Comparator: NGO-delivered youth outreach programmes serving troubled youth | Healthcare, plus time of participants and in-kind support Not stated International Dollars ($Int) | CUA 6 months | Transformation of WHO-DAS II scores to generate QALYs. Mean gain of 0.006 QALYs | Total mean costs per participant for intervention were $Int 104.1, including $Int 82.1 in monetary costs | Incremental cost per QALY gained $Int 7,260. Authors assume unlikely to be cost-effective in Sierra Leone |

*(Continued)*

Table 1. (Continued)

| References Country of study | Setting and study population (age, sex, size) | Intervention details (study design, description of intervention and comparator) | Economic analysis | | Outcomes and key findings | | |
|---|---|---|---|---|---|---|---|
| | | | Perspective Price Year Currency | Type of economic analysis Study duration | Effect on mental health | Resource and cost impacts | Key economic findings |
| Park et al., 2022 Turkiye | 627 adults with mild psychological distress (scores between 3 and 12 on GHQ but no diagnosed disorder) | RCT. Intervention: 5-session, group-based, stress management course in which participants learned self-help skills. Includes illustrative book plus enhanced usual care. Comparator: enhanced usual care | Health and social care system 2019 Turkish lira (TL) | CUA 1 year | Significant improvement in QALYs over 6 months: 0.42 versus 0.39 ($p = 0.001$) | Mean total costs in intervention group, including intervention TL214.49 versus TL20.32 ($p = 0.001$) | Incremental cost per QALY gained TL6,068, well within reported cost-effectiveness thresholds in Turkey |
| Priebe et al., 2010 Serbia, Croatia and Bosnia-Herzegovina | 526 adults with a diagnosis of war-related PTSD using the Clinician-Administered PTSD Scale for DSM-IV (CAPS) | Observational before and after study for adults starting to receive unspecified varied treatment for PTSD at specialist treatment centres | Health and social care system 2005 Euros (€) | CCA 1 year | 14% of the cohort did not have PTSD after 12 months. There was a small but significant change in PTSD symptoms on CAPS ($p < 0.001$) | Annual mean costs, including treatment ranged between €128 in Bosnia to €382 in Serbia | Mean 12-month costs for people with PTSD at 12 months €307 ± €260. Mean costs without PTSD €284 ± €218 (Non-significant) |
| Rohr et al., 2021 Germany | 133 Syrian refugees aged 18–65 with mild to moderate posttraumatic stress symptom severity (score of 11–59) on the Posttraumatic Diagnostic Scale for DSM-5 (PDS-5) | RCT: Intervention: smartphone app providing cognitive behavioural therapy-based self-help. Comparator: Psychoeducational reading material | Health care payer 2019 Euros (€) | CUA 4 months | There was no significant difference in mean QALYs at follow-up (0.290 versus 0.294) $p = 0.35$ | Total mean intervention and health care costs were non-significantly lower in intervention group €384 versus €484 $p = 0.38$ | The intervention was very unlikely to be cost-effective. Only a 20% chance at a plausible cost per QALY threshold of €50,000 |

Abbreviations: CBA, cost-benefit analysis; CBT, cognitive behavioural therapy; CCA, cost consequences analysis; CEA, cost-effectiveness analysis; CRIES-8, Child Revised Impact of Events Scale; CUA, cost-utility analysis; DSM-IV, Diagnostic and Statistical Manual of Mental Disorders, Fourth Edition; GHQ, General Health Questionnaire; PTSD, post-traumatic stress disorder; QALYs, quality-adjusted life years; RCT, randomised controlled trial; ROI, return on investment analysis; WHO-DAS 2.0, World Health Organization Disability Assessment Schedule 2.0.

moderate depression in newly arrived refugees in Germany would be €19,779, compared to case-finding via self-referral followed by psychosocial care (Biddle et al., 2019). Although no formal QALY decision-making threshold exists in Germany, the authors concluded screening may be a cost-effective strategy.

Evaluations often looked at brief psychological interventions. One intervention study is an economic evaluation conducted alongside a randomised clinical trial (RCT) looking at the case for Self-Help Plus (SH+), a group-based, guided, self-help psychological intervention for 627 Syrian refugees in Turkey (Park et al., 2022). It is intended to increase the stress-management capacity of adults exposed to adversity, reduce psychological distress, and prevent the onset of mental disorders. Compared to enhanced usual care alone, at 1 year follow-up the SH+ group had significantly improved quality of life, with a cost per QALY gained of TL 6,068, a value considered to be cost-effective in a Turkish context. In contrast, another RCT of smartphone-delivered cognitive behavioural therapy-based self-help intervention for Syrian refugees with

PTSD only had a 20% chance of being cost-effective at 4-month follow-up in Germany at a notional cost per QALY threshold of €50,000 (Rohr et al., 2021).

The economic case for a brief multi-component psychological intervention designed to address common mental health problems in conflict-affected settings, Problem Management Plus (PM+) therapy, has been evaluated for 340 adults in a RCT of conflict-afflicted regions of Pakistan (Hamdani et al., 2020). At 3-month follow-up anxiety and depression symptoms and psychosocial functioning were significantly improved, compared to enhanced usual care only, with no differences in health service utilisation. Overall, PM+ was more effective but more costly. However, as only incremental costs for different clinical outcomes were reported, rather than QALYs or monetary CBA, policy makers cannot easily judge if this is a cost-effective use of resources. The results might also have been more conclusive different if data on employment and informal care impacts collected were included in the economic analysis.

PM+ for Syrian refugees in both the Middle East and Europe is also being examined (Sijbrandij et al., 2017). A pilot RCT on the cost-effectiveness of PM+ plus usual care for 60 Syrian refugees with elevated psychological distress and reduced psychosocial functioning in the Netherlands was compared to usual care only (de Graaff et al., 2020). Significant improvements in levels of depression, anxiety and social functioning were found in the PM+ group. While health and social service utilisation and costs were higher, productivity losses due to time out of employment or education, and need for family informal care were lower. Overall better outcomes were reported with no significant difference seen in costs, with an incremental cost per recovery from anxiety and depression of €5,047 and €2,266 from health system and societal perspectives. While encouraging, to be more definitive the main trial and economic evaluation in the Netherlands is also measuring changes in QALYs in order to estimate the cost per QALY, as well as looking at changes in health and productivity over a longer 1-year time period.

In Sierra Leone an economic evaluation alongside a RCT looked at the short-term cost-effectiveness of the Youth Readiness Intervention (YRI), a 10-week multi-component therapy to reduce functional impairment from depression, anxiety and other emotional problems in conflict-affected young people (McBain et al., 2016). Only very marginal improvements in QALYs were observed meaning the cost per QALY gained was not deemed to be cost-effective, but the economic analysis may have been different if the benefits to education observed in study, including significantly higher continued rates of school enrolment, attendance, as well as academic performance including completion of coursework, classroom behaviour and classroom participation, had been taken into account.

A RCT involving 154 conflict-affected children in the Palestine Territory prospectively collected information on the costs of a brief 5 session group-based cognitive intervention to address PTSD. Significant reductions in PTSD post-intervention were reported, at a cost per participant of $39, a value the authors considered to be low, but no formal assessment of cost-effectiveness was made, nor any collection of data on changes in health and other service use (Barron et al., 2016).

Models of stepped care where the most intensive intervention support is only provided to those with the greatest levels of enduring mental health need may prove potentially attractive, as they have less resource impact than strategies where all participants receive all levels of care. A four-level strategy in Germany involving watchful waiting, peer or phone support, group and then finally individual psychological therapies have been evaluated in a trial of almost 600 Arabic or Farsi-speaking refugees and asylum seekers with at least mild depressive symptoms (Böge et al., 2022). Subsequent health service costs over a 1-year period were significantly lower in the intervention group, with improvements in depression also seen. The strategy appeared to be highly likely to be cost-effective in sensitivity analyses with lower overall costs and slightly better QALY outcomes than care as usual.

The importance of looking at the long-term impacts, as well as the wider benefits of intervention for families and communities, can be seen in an economic evaluation as part of an observational study of traumatised refugees living in Denmark who have been victims of torture (Bager et al., 2018). Using registry data the study compared the long-term employment and health service utilisation costs over 14 years for 44 refugees who received multi-disciplinary rehabilitation with a matched group of refugees on a wait list for treatment. While benefits did not outweigh costs when looking at impacts to

individuals alone, once impacts on immediate families were also considered the intervention generated a positive net benefit after 3 years.

A modelling study used pilot RCT data on a multidisciplinary rehabilitation programme including physiotherapy, biofeedback-supported psychotherapy, such as relaxation techniques, and social support for 34 people in Kosovo with trauma, anxiety or depression who had been victims of torture or witnessed the execution of a family member during the Kosovan war between 1998 and 1999 (Chang et al., 2018). While limited by small-scale and lack of data on changes in health and social service resource use, even when making the assumption that improvements in quality of life seen at 3 months could be sustained for 1 year, the authors concluded that the cost per QALY gained of €10,505 would not be considered cost-effective, as this was more than three times GDP per capita in Kosovo. However, if self-reported improvements in participants' income over 3 months were sustained for 57 months there would be a positive ROI. Another evaluation of the impact of receiving specialist treatment on a cohort of adults with war-related PTSD in three other countries of the former Yugoslavia was unable to find any difference in the costs of health service utilisation, including intervention costs, between those with or without PTSD at 1 year follow-up (Priebe et al., 2010).

## Facilitating the future use of economic evaluation

The scoping review revealed 11 economic evaluations focusing on actions for refugees and displaced people. Four of these were reported to be cost-effective within their settings (Bager et al., 2018; Biddle et al., 2019; Böge et al., 2022; Park et al., 2022), with another three having improvements in outcomes at potentially acceptable additional costs (Barron et al., 2016; de Graaff et al., 2020; Hamdani et al., 2020). Of the remaining four, a smartphone-delivered self-help intervention in Germany did not have an impact on quality of life and therefore was not cost-effective (Rohr et al., 2021). A multi-component therapy for conflict-affected youth in Sierra Leone did not appear to be cost-effective from a health system perspective, but potentially might be cost-effective if benefits to the education sector were also considered (McBain et al., 2016). A very small modelling study in Kosovo suggested an eventual positive ROI nearly 6 years after intervention (Chang et al., 2018). Another observational uncontrolled study comparing costs and consequences of unspecified interventions for PTSD reported marginally better outcomes at no extra cost (Priebe et al., 2010).

While these studies are broadly positive this evidence base remains very limited. We now discuss ways in which the evidence can be strengthened to help facilitate the use of economic evidence in decision-making processes. These include more consistent collection of data on quality of life as part of effectiveness evaluations, as well as looking at impacts over a longer-term time periods of at least several years, so that more potential benefits of early intervention can be identified. The perspective adopted in economic evaluations may also need broadening to include intersectoral benefits beyond health, as well as identifying complementary benefits to host communities, given the political reality in some settings that policy makers will prioritise investment in measures supporting their domestic populations, regardless of their cost-effectiveness or moral imperative. The costs of implementation also need to be carefully measured; costs may be less of a budgetary concern if interventions can be effectively delivered online or facilitated by non-specialists or refugee peers. More use can be also made of

modelling, drawing on evidence on the effectiveness and resource requirement of interventions delivered in comparable settings. The budgetary impact of any proposed strategy should be considered; modelling could also be used to look at how implementation might be adapted to contain some of these costs. Finally, economic evaluation is a tool that should not be used in isolation, it is also important to collect information on the contexts in which interventions are implemented, in order to identify factors that can influence cost-effectiveness. This contextual information can also help in assessing the case for intervention adaptation, replication and scale-up.

### Collect data on quality of life

Seven of these 11 evaluations estimated changes in quality of life as their primary economic outcome measure, making it easier for decision-makers to compare their cost-effectiveness to that of other alternative health care interventions. Ensuring that future effectiveness studies collect quality of life data as one of their outcomes, even if an economic evaluation is not planned, will also allow this evidence base to expand further, as these data could be used to inform modelling studies. It may also be possible to retrospectively conduct economic evaluation by synthesising available evidence from trials on changes in quality of life, along with intervention costs and changes in patterns of service utilisation.

### Assess short, mid and long-term economic impacts

Some of the benefits of investing in better mental health are immediate, but many will accrue over many years. Quantifying the immediate (1 year), mid-term (up to 5 years) and the longer-term (beyond 5 years) economic impacts of intervention may also help strengthen the case for investment. Many of the economic benefits of better mental health are though associated with mid to long-term benefits to society such as improved educational attainment and participation in work, reduced contact with criminal justice, as well as better physical health; all of this in turn will reduce reliance on welfare benefits.

Health economic arguments on their own may not be sufficient to persuade decision-makers to think long-term. Public health policies that recognise the importance of taking a long-term approach to health will also be critical. That said, short and mid-term health and economic impacts of any investment should also not be ignored, as these can be helpful in providing an initial justification for investment, particularly in settings where electoral cycles are short. In the case of refugees and displaced people, we have noted that the lack of early intervention has been argued to be potentially a critical factor in long-term higher health system utilisation and societal costs (Bauhoff and Göpffarth, 2018; Blackmore et al., 2020; Tzogiou et al., 2022). A regression analysis covering 17 countries has also observed an association between increased overall spending on refugees in general and better mortality outcomes (Tan et al., 2016). Factors such as investment in fair protection and documentation processes, healthcare services, logistics and support, as well as external relations were linked with better outcomes. However, with the exception of one study (Bager et al., 2018), none of the economic evaluations in Table 1 look at the economic case beyond 15 months. Ideally prospectively collecting data on costs and outcomes over several years as part of evaluations would allow for assessment of short, mid and long-term impacts, but this practically may be difficult to do.

One feasible alternative is to make more use of economic decision modelling approaches, frequently used in health economic evaluation (McDaid, 2014), to draw on multiple sources of existing data, including short-term trial outcomes, as well as health and welfare system registries where available, as used in the modelling study in Denmark (Bager et al., 2018), and/or previous longitudinal analyses following health service use of refugees, to estimate mid and long-term costs and outcomes, potentially over decades. Sensitivity analyses in models could also be used to deal with uncertainty by varying assumptions on model parameters, as well as looking at how the economic case might change for different plausible scenarios, for instance, different levels of intervention uptake, likely long-term effectiveness, differences in access to routine health care services, or even on the level of time that refugees will spend in any host country.

### Ensure economic evaluations consider impacts that are of relevance to funders

Economic evaluations need to take account of outcomes and goals that are of particular relevance to their funders. This can be complex, as there may be entirely separate and uncertain funding streams for refugee health services even in HICs, as in Germany where responsibility for funding services is fragmented between different administrative bodies in different regions (Biddle et al., 2022). Many LMICs rely on what can be uncertain and time-limited levels of external donor aid from HICs, as well as direct financial support from the United Nations High Commissioner for Refugees (UNHCR) and other international agencies (Spiegel et al., 2018). There will also be a reliance on non-governmental organisations to deliver and in many cases fund services.

The direct role of LMICs in using their own resources to funding refugee health services remains limited, although if refugees are allowed to work they may then contribute to national health (or other) insurance schemes, as has been the case in Iran for Afghan refugees (Spiegel et al., 2018). Some longer-term funding from international agencies is conditional on the provision of specific services, as for instance is the case with the Global Concessional Financing Facility, a scheme that has provided access to cheap long-term loans as well as grants to support health and wider welfare of both refugees and host communities to LMICs, including Lebanon (Global Concessional Financing Facility, 2022).

The objectives of these funders may be much broader than mental health; identifying and valuing any additional benefits of intervention for physical health and general functioning of target populations may be important, given that funders may have to balance the physical health as well as mental health needs of displaced populations. Some may be interested in wider issues of economic development. Donor organisations may also be more interested in determining how best they can use their limited resources to support the victims of different geo-political crises, so it may be useful to compare the relative cost-effectiveness of interventions in different settings, or consider how they could be adapted to be used in other settings.

External funders may wish also to look at the cost-effectiveness of health system versus non-health system interventions such as welfare/development assistance; for instance organisations such as the World Bank have advocated for more use of cash transfers programmes (Özler, 2017) and there is a growing body of research looking at whether interventions that help alleviate poverty, including pandemic-related impacts and/or encourage young people to remain in education may be more effective and cost-effective in

improving mental health compared to psychological interventions alone (Bauer et al., 2021; Zimmerman et al., 2021; Zaneva et al., 2022).

### Broaden economic evaluations beyond health outcomes to look at impacts that are of key interest to host country policy makers

There are further challenges in making any economic case to local policy makers to fund services that are not targeted at their local populations. In the case of refugees and asylum seekers, there may be considerable uncertainty on how long they will remain in the host country; they may be able to return home or alternatively may wish to transit to a final destination country. If they are likely to move on to another country, policy makers may feel that any long-term mental health and wider socio-economic benefits will be gained in these other countries rather than their own. Policy makers, particularly in LMICs, may also feel that any support for externally displaced persons means that they are having divert their own very limited mental health resources away from their own population's health. These concerns over pressure on health systems can also be visible in HICs, who may be reluctant to even receive refugees, despite their legal obligations. Domestic politics might also play a role, with a reluctance of some politicians to be seen to be funding services for refugees (Biddle et al., 2022).

All of these factors may mean that in designing economic evaluations, particularly in low-resource settings, it is important to assess not only the potential costs and benefits to the intervention target population, but to also look at potential wider impacts in the short, as well as long-term, for the local population. Modelling analyses could be used to estimate the economic benefit for extending coverage of an effective intervention to the local population; for instance, policy makers in Lebanon may be more willing to endorse the provision of brief psychological interventions for refugees if there is evidence that these would also have benefits for the mental health and life chances, for example, participation in work or education, of the local population affected by the Beirut port explosions and the ongoing economic crisis (Farran, 2021).

In HICs similarly identifying other population groups that might benefit from similar interventions could include military veterans and aid workers who have also been exposed to conflict-related trauma. This may also help to sustain the capacity and maintain skills for delivering interventions, as there will be fluctuations over time in the need to support refugees and internally displaced people. Psychological and other interventions targeted at refugees are now being adapted to support the mental health of high-risk population groups during the COVID-19 and future pandemics, for example, health and social care workers (Hooper et al., 2021; Ottisova et al., 2022).

### Explore how costs of implementation may be contained

The costs associated with effective implementation also need to be carefully measured; these include any potential resource requirements and costs for cultural adaptation, as well as translation/interpretation for interventions, and initial training for service deliverers. There may however be ways in which these costs could be reduced and thereby strengthen the economic case for investment. Economic modelling analyses could be used to look at how cost-effectiveness changes using different delivery mechanisms, comparing online versus face-to-face interventions, or ongoing training and supervision provided by local rather than external experts.

There may also be considerable differences in budgetary impact depending on who is delivering any mental health intervention; task-shifting away from mental health specialists so that interventions are facilitated in primary care potentially by lay community health workers or refugee peers may reduce costs considerably. It may also have an impact on the level of uptake of services, for instance, some refugees may prefer to be supported by someone from their country, who shares a common language and culture.

### Placing economic evaluation within a broader systems context

It is important not to see economic evaluation as a tool that should be used in isolation, but one that can be used alongside other sources of evidence. We have noted broad eco-system challenges that refugees and other displaced people face in host countries. In many settings, including but not only LMICs, there may be substantial barriers to mental (and other) health service access for refugees and displaced people, even where these countries are considered to have some form of the universal health coverage and have integrated some services (Spiegel et al., 2018; Satinsky et al., 2019; Pollard and Howard, 2021). Factors can include the overall limited provision of mental health services, high levels of out-of-pocket payments, lack of cultural adaptation and trained staff, as well as stigma around mental health.

One qualitative systematic review focused on how context influenced implementation of non-pharmacological interventions for refugee mental health in LMICS. This identified that interventions that provided work or education opportunities for refugees were received favourably. It also emphasised the importance of the cultural context on intervention delivery (Jannesari et al., 2021). Other contextual factors that can have an impact long term mental health recovery and related economic outcomes in both permanent and transitional host countries, include the legal rules in place concerning access to employment and education (Juárez et al., 2019; Posselt et al., 2019).

This means it is important to undertake process evaluations to better understand the implementation process, as well as situational analyses to better understand the contexts in which mental health interventions are being delivered. These wider types of information were not considered in the economic evaluations we have identified, although one ongoing study is making use of exactly this type of data in making the economic case for scaling up access to mental health services for Syrian refugees in Europe and the Middle East (Sijbrandij et al., 2017).

An intervention may appear cost-ineffective because of wider contextual factors in a setting, for instance, the cost and time needed to reach a service, or perhaps because of long-term uncertainty on achieving settled legal status. If these adverse factors can be identified, then they can potentially be taken into account in economic analysis, for instance modelling the costs and benefits of additional measures to overcome implementation issues. Examples could include additional investment in cultural adaptation, additional training, language support and the use of lower-cost interventions, for example, peer delivery. In the same way, the impacts of legislative measures that allow for refugee employment and/or the ability to be covered by national health insurance systems could be considered.

## Conclusions

There is not only a moral case to protect and support global mental health, there is also an economic case (Chisholm et al., 2016). However, expenditure on mental health remains low in many countries, being 1.5% or less of total governmental health expenditure in LMICs. This implies that more needs to be done to increase awareness of this economic case, not only within countries but also across international agencies and NGOs, so as to promote the use of lower-cost evidence-based interventions that can be delivered in non-specialist settings, such as those set out in the WHO's mhGAP Intervention Guide (Keynejad et al., 2021).

Raising the profile of mental health in LMICs and strengthening mental health capacity generally within countries will also help facilitate better access to services for refugees and displaced people, recognising nonetheless that additional training for service providers may be needed and services culturally adapted. Investment in refugee mental health will also be aided through economic evidence. We have argued here, however, that the mental health of refugees and displaced people is an area of global mental health where the economic evidence appears to be very limited, with our rapid scoping review only revealing 11 economic evaluations. We searched through eight databases, including some focused on LMICs, with no language restrictions. Although we may have missed some studies not catalogued in these databases, our findings are in line with other studies. One previous systematic review of economic evaluations for PTSD for all populations found just one study on refugees (von der Warth et al., 2020) while a systematic review of effectiveness studies on mental health interventions for conflict-affected populations found no economic evaluations at all (Al-Tamimi and Leavey, 2022).

To help build the evidence base, evaluations should as standard, and in addition to assessing effectiveness, record all resources required to deliver interventions so that the costs of implementation and therefore budgetary implications can be estimated. This will also allow programme implementers and policy makers to consider whether resources need to be adjusted if interventions are scaled up or replicated in different country settings. When designing evaluations, where practical, a measure that can be applied to all health conditions, such as the QALY, should also be considered, as this will aid decision-makers in making comparisons between different health interventions.

The duration of studies may also make a difference, the longer the evaluation the more likely that it can account for changes in health and other service utilisation, as well as social functioning, including work participation, that may not be evident in studies of less than 1 year. This may also help strengthen the argument for investment. In the longer term, it is important to build economic evaluation as standard into evaluation plans and study protocols; a number of protocols for ongoing studies on refugee mental health have done this (Sijbrandij et al., 2017; Purgato et al., 2019; Durbeej et al., 2021; Weise et al., 2021). This also implies further building local capacity to undertake and interpret the results of economic evaluation in LMICs.

In the short term, it may also be feasible to retrospectively make use of effectiveness data on relevant mental health interventions and model their cost-effectiveness. The potential economic benefits of replicating an intervention shown as cost-effective in one country setting to another similar country could also be modelled. Models could also be used to extrapolate short-term findings over longer time periods and to show how different levels of uptake and sustained engagement impact on outcomes. Contextual factors, such as differential access to culturally appropriate health and welfare services, as well as the right to employment, could be documented and also considered when interpreting the results of any economic evaluation, as well as taken into account when using modelling.

Refugees, in particular, may not be seen as an asset in host countries, where there are many competing demands to improve the health and well-being of the domestic population. Economic arguments in host countries critically may need to also demonstrate that improving the mental health of refugees potentially has benefits for the local population, such as services being expanded to other groups that have gone through traumatic experiences, including health and other frontline workers during the COVID pandemic. This is in addition to including intersectoral costs and benefits beyond the health sector given that the funding and delivery of many refugee mental health-related services may be fragmented. In cases where services are funded by external donors, evaluations need to also be mindful of the overall goals of these donors. External donors might also take steps to encourage funding of integrated services that also benefit host populations. The need for services for refugees and internally displaced people has been increasing in recent years, and may well increase further not just because of conflicts, but also due to economic hardship and natural disasters. Economic evaluation is therefore an additional tool complementing the already compelling arguments for further investment in global mental health.

**Open peer review.** To view the open peer review materials for this article, please visit http://doi.org/10.1017/gmh.2023.1.

**Supplementary material.** To view supplementary material for this article, please visit https://doi.org/10.1017/gmh.2023.1.

**Data availability statement.** The search strategies used for the bibliographic databases in the scoping review are available as Supplementary Material.

**Author contributions.** D.M.D. conceived the focus of the manuscript, undertook the scoping review and wrote the first draft. A.L.P. contributed to the scoping review, contributed to the manuscript design and critically revised the manuscript.

**Financial support.** This research received no specific grant from any funding agency, commercial or not-for-profit sectors.

**Competing interest.** The authors do not have any direct conflicts of interest, but are authors of two of the economic evaluations included in the scoping review.

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
