## [Reviewer Report]

Dr Dixon Chibanda, Dr Judy Bass, 

Co-editors in Chief,

Dr Marit Sijbrandij, 

Senior Editor

Cambridge Prisms: Global Mental Health

14 August 2022

Dear Professors Chibanda, Bass and Sijbrandij,

In response to your invitation, I have on behalf of myself and my co-author, A-La Park, submitted our manuscript entitled ‘Making an economic argument for investment in global mental health: the case of conflict-affected refugees and displaced people’ to Cambridge Prisms: Global Mental Health for your consideration. 

The initial invitation was to look broadly at the cost effectiveness of all interventions in global mental health settings, which is a very broad topic indeed. Taking this as our starting point we have drafted an overview review which sets out some key issues to be considered when undertaking an economic evaluation of any global mental health intervention, but illustrating this by focusing on the area of conflict-affected refugees and internally displaced people. 

We have specifically chosen this focus as it is an area of global mental health that is truly global as many high-income, as well as low and middle-income countries, receive considerable numbers of refugees, and the numbers appear only to be growing in recent years. The evidence base on the adverse long term health impacts of poor mental health in refugees and displaced people is considerable and this also includes some evidence on the long-term profound consequences of neglected mental health, not just for health but for life chances in these populations, which we have highlighted. Yet it appears to be an area when there have been few economic evaluations to date, and we believe there is considerable scope for strengthening the evidence base. 

In the broad scoping review we have undertaken we identified very few economic studies looking at the case for mental health of refugees and displaced people. We highlight this evidence and its limitations and make recommendations on how it can be expanded. This can be contrasted with the much more developed economic evidence base that we identified for screening, vaccination against and treatment of communicable disease in refugees. In this case, it is perhaps easier for policy makers, to make the domestic case for action, as conceivably there may be immediate protective benefits to the domestic population. 

This is an example of a particular and almost unique challenge in health economics, namely that in the case of refugees and to some extent internally displaced people, they have to rely on the goodwill of external funders for access to services. Not all policy makers will see the mental health needs of these groups as a high priority, especially if resources for their domestic populations are very tight. They may not be able to see what benefit they domestically from investment. 

This is why we call for more focus in future research and evaluation not only on more focus on the long-term benefits of better mental health for refugees and displaced people but also for more consideration in evaluation of any potential additional positive impacts in host countries of investment for their domestic populations. These benefits for instance could include extending access or replicating new services to support local people with similar psychological needs, e.g. health and other frontline workers affected by the pandemic or internal conflicts or hardships such as natural disasters. 

On behalf of myself and my co-author, we sincerely hope that you find our overview review to be helpful and of interest to Cambridge Prisms: Global Mental Health and look forward to hearing from you.

Yours sincerely

David McDaid

Associate Professor in Health Economics and Health Policy,

Care Policy and Evaluation Centre, Department of Health Policy

London School of Economics and Political Science

London, UK.

E-mails: d.mcdaid@lse.ac.uk

---

## [Reviewer Report]

*Comments to Author*: This paper seeks to identify economic evidence available to build an economic case investment for delivering mental health intervention to the refugee population. The author focuses on three major basic arguments: costs of not taking action, costs of investments needed and economic evaluation available in English published literature. Eleven economic evaluations were found with mixed results in terms of the cost-effectiveness of such interventions. The author raised some methodological issues about the limitations of the economic evaluation findings and recommended some potential solutions for overcoming the gap in research studies on this topic. The conclusion is that more than a moral argument, there are potential benefits of investing in this group for countries’ economies.

The topic of this paper is relevant for global mental health not only due to the high prevalence of displaced people in the world but mainly because of the short and long-term deleterious consequences for the health system, and social and economic sectors. Of note, the majority of refugee people are in low and middle-income countries. In this regard, the method used in this paper searching only English-published languages does not reflect the challenges and pitfalls of LMIC context to build an economic argument according to the author’s goals. The majority of literature regarding LMIC falls out of the three databases used in this paper. Most of the economic studies evaluated in this paper are from European countries. I missed some discussion about the best arguments that can be built in a very limited budget constraint scenario, with a high level of poverty, unemployment, inequity and frequently, corruption and inefficiency of resource use.

It is widely known that mental illness is closely influenced by socio-determinants factors. In this regard, refugees face, at least, two major sets of factors: the first group is related to the reason for displacement (war, political persecution, etc.) and all related stressors with this acute situation. The second group is related to how welcomed and supported they are by the country host. This latter ultimately determines the level of economic, social and health vulnerability they live. Acute intervention is related to conflict consequences, new country adaptation, and length of time to have access to the health system. This varies from one country to another and in this regard, it would be nice to see how these factors influence mental health outcomes. The way people are welcomed and inserted into society may exert a significant influence on adherence, access and treatment effectiveness. The suicide rate is important too. It is important to keep in mind that in this population environmental factors are crucial to be considered in economic evaluation and mental health status. The mental health treatment gap is a reality for two-thirds of the general population but refugees may need a more comprehensive approach taking into account these factors.

There is vast literature on this topic regarding the scenario of LMIC. For instance, the type of health system (universal coverage or out-of-pocket), a social insurance scheme for vulnerable populations, and how refugees are included in the country (or not!). The higher level of vulnerability, the poor mental health outcomes and response to treatment. These elements are important not only for perpetuating poor mental health but also for some crucial steps in the cost-effectiveness of mental health interventions. For instance, the short length of time to obtain mental health treatment is an important factor for better outcomes. Then, access to healthcare systems is a bottleneck point. Other issues are related to the preparedness of the health providers to receive this population (language barriers, cultural issues, stigma, lack of training in some interventions, etc.). It would be interesting to identify the characteristics of the cultural, social and geographic context in which such positive results of these economic evaluations were observed.

Yet, the treatment offered to a highly vulnerable population should be integrated into a set of complex actions regarding poverty, unemployment, hunger, violence, cultural values, stigma and others. It is difficult to consider that the intervention is not cost-effective if these variables are not controlled at some point. Refugees are not a single and homogeneous variable and the environment is extremely important. Also, considering the long-term consequences, the high stress and vulnerability level the refugees face is dangerous for their subsequent generations too, leading to higher mental health conditions and costs in the future. The main problem of mental health conditions and chronic vulnerability status is that actions should be sustainable for the long term and their effects may be slow and invisible to policymakers focused on seeking short-term results. In a very limited budget scenario and with multiple urgent population needs, policymakers are not willing to invest in the long-term actions even recognising its importance. The policymaker willing-to-invest in a not voter group in the long term is another important bottleneck. A moral argument may be useful for short-term actions but it is important to have a public health policy addressed to the long-term goals and this is not frequent in many LMIC. In this regard, NGOs exert an important role to pressure governments in favour of such groups. Some of them deliver some mental health interventions not necessarily based on scientific evidence and there is a gap of evidence on the effectiveness of such organisations in this regard. Many LMIC don’t draw health policies based on scientific evidence and, yet, mental health treatment is not offered for the entire population in need. For instance, there is a treatment gap for depression and anxiety of 60-70% in the Latin America continent, despite the high prevalence, costs and burden to the society in the region. Then, the puzzling argument remains despite economic evaluation findings and reasonable economic arguments. It may be interesting that aspects such as equity, poverty, and vulnerability are aligned with economic arguments because some countries are more inclined to equity issues while others are driven by cost issues.

Another issue in this paper is the methodology used in economic evaluation. Cost-utility studies don’t include social and indirect costs in the analysis and QALY poorly captures the real benefits of mental health interventions, especially in this group of multiple needs. I agree that broadening economic evaluation is crucial in this regard as pointed out by the author.

---

## [Reviewer Report]

*Comments to Author*: The topic that the manuscript “Making an economic argument for investment in global mental health: the case of conflict-affected refugees and displaced people ” is very important. The authors call for more health economic studies evaluating interventions for this vulnerable group aimed to improve mental health and avoid very costly undesirable outcomes for individuals and society .

I have only few comments and questions.

1. In the “Impact statement” – line 15-16: “in contrast we identified very few economic studies looking at the case for mental health” – to be precise, few economic studies looking at the case for mental health for refugees, asylum seekers and other displaced people

2. In the abstract – line 39 “population group” – better to name the population group, to keep the focus on refugees, asylum seekers and other displaced people

3. In the abstract - line 40 : “All but two of these studies potentially could be cost effective, but only five studies reported cost” – change “studies” to “interventions” and add “studies” after “five

4. In the Introduction – can you also present the percentage of mental healthcare budget of the total healthcare budget (line 68), at least for HIC?

5. Page 9 line 207 – 208: “In many HICs economic evaluation is an integral part of the policy-making process within health systems” – I am not totally agree. Economic evaluations are mainly used while approving new drugs, not new “non – pharma” interventions

6. Page 11 line 289 – spelling check “We alo

7. Page 12, line 96-97 “economic case for mental health screening” – for this particular population?

8. Page 12 line 303-304 “One intervention (study?) for 627 304 Syrian refugees is an economic evaluation conducted alongside a randomised clinical trial … - unclear sentence

9. Page 14, line 281 “cost per QALY gained of €10,505 would not be considered cost-effective” – why not cost-effective? Threshold?

10. Conclusion – the first paragraph might be shorter, this is very general.

11. Table. Chang et al 2018 – “Cost per QALY gained between intervention and control 0.097.” You mean difference in QALY?